# Heteroscedastic uncertainty quantification in Physics-Informed Neural Networks

**Olivier Claessen & Yuliya Shapovalova & Tom Heskes**
Institute for Computing and Information Sciences, Faculty of Science
Radboud University
6500 GL, Nijmegen, Netherlands
{Olivier.Claessen, Yuliya.Shapovalova, Tom.Heskes}@ru.nl

## Abstract

Physics-informed neural networks (PINNs) provide a machine learning framework to solve differential equations. However, PINNs do not inherently consider measurement noise or model uncertainty. In this paper, we propose the UQ-PINN which is an extension of the PINN with additional outputs to approximate noise. The multi-output architecture enables the approximation of the mean and standard deviation in data using negative Gaussian log-likelihood loss. The performance of the UQ-PINN is demonstrated on the Poisson equation with additive noise.

## 1 Introduction

Alternative approaches to approximate partial differential equation (PDE) solutions have emerged with advances in scientific machine learning (SciML). One of the methods that recently emerged in SciML is Physics-Informed Neural Networks (PINNs) (Lu et al., 2021). Using this approach, physical constraints are encoded as additional loss functions, where information about the physical system is used as a regularizer when fitting the model (Raissi et al., 2017; Karniadakis et al., 2021). PINNs, also called vanilla PINNs, typically utilize a mean squared error loss function and cannot quantify uncertainty.

Several approaches have been proposed to enable uncertainty quantification in PINNs. These include negative Gaussian log-likelihood as a loss function and a single scalar, which is used to approximate additive noise in the data set Xiang et al. (2021). In addition, Bayesian neural networks in combination with sampling methods such as Monte Carlo or Variational Inference Sun & Wang (2020); Dandekar et al. (2020); Linka et al. (2022); Zou et al. (2022). Another approach is to create an ensemble of models such as PINNs Zou et al. (2022); Psaros et al. (2023). However, many of these types of methods are generally computationally expensive and do not always account for heteroscedastic noise in the data.

We propose a multi-output PINN architecture called the UQ-PINN that can quantify aleatoric uncertainty and account for measurement noise. This approach is computationally inexpensive and can closely approximate the standard deviation for different types of homoscedastic and heteroscedastic noise. We evaluate the method on the Poisson equation corrupted with additive noise and compare its performance with a regular PINN.

## 2 Methods

### 2.1 Physics-informed neural networks

Given a system of parameterized nonlinear partial differential equations (PDEs) of the form

$$\mathcal{F}[u(x)] = f(x), \quad x \in \Omega; \quad \mathcal{B}[u(x)] = b(x), \quad x \in \Gamma, \tag{1}$$

where $x$ is the $D$ dimensional space or spacetime coordinate vector defined in the physical domain $\Omega \subset \mathbb{R}^D$ with boundary $\Gamma$. $\mathcal{B}(\cdot)$ is a boundary condition operator and $b(x)$ are Dirichlet boundary conditions. $\mathcal{F}(\cdot)$ is a differential operator and $f(x)$ is typically referred to as the source term or

driving force. The goal of a PINN is to find a surrogate model $\hat{u}(x, \theta)$ that approximates the unknown PDE solution $u(x)$ of this PDE system. The PINN $\hat{u}(x, \theta)$ is parameterized by $\theta$, the parameters of the neural network. In the remainder of the paper, the dependence of $\hat{u}$ on $\theta$ is omitted for simplicity.

We focus on forward-deterministic PDE problems, in which the operators $\mathcal{F}$ and $\mathcal{B}$ are known. For the source terms $f(x)$ and the boundary conditions $b(x)$, we have access to noisy measurements at a finite number of (collocation) points in the domain. Specifically, the observations $\tilde{f}_i$ of the actual source term and measurements at the boundary conditions are corrupted with noise:

$$\tilde{f}_i = f(x_i^{(f)}) + \epsilon_f , \quad \tilde{b}_i = b(x_i^{(b)}) + \epsilon_b , \tag{2}$$

where $\epsilon_f \sim \mathcal{N}(0, \sigma_{(f)}^2(x_i^{(f)}))$ and $\epsilon_b \sim \mathcal{N}(0, \sigma_{(b)}^2(x_i^{(b)}))$ are Gaussian noise terms with heteroscedastic variance. Noise on the source term can be interpreted as a lack of knowledge of the exact parameters of the PDE. Noise at the boundary conditions can be a direct result of noisy sensor measurements. Measurements at the $N_f$ collocation points have been sampled $S_f$ times, giving a data set $D^{(f)}$ of $N_f \times S_f$ data points. The boundary condition data set $D^{(b)}$ contains $N_b \times S_b$ data points.

## 2.2 UQ-PINN

To quantify aleatoric uncertainty, we propose to extend the PINN with additional outputs to predict the standard deviations $\hat{\sigma}_f$ and $\hat{\sigma}_b$ as a function of input coordinates $x$, we call this extension UQ-PINN. In our problem set-up, we allow for additive noise both in the source terms and on the boundary condition measurements. Our goal is not only to estimate the PDE solution but also the standard deviations to help quantify the uncertainty. We use feedforward neural networks with the hyperbolic tangent activation function in all hidden layers. The output layer uses linear activation for the estimation of the mean and exponential activation for estimation of the standard deviation to ensure positive values.

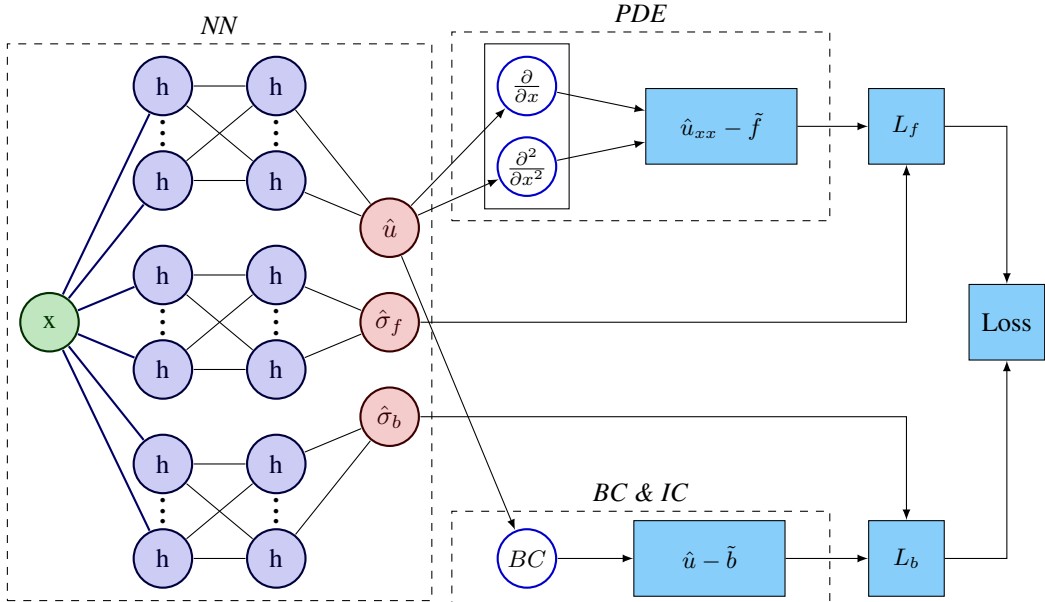

Figure 1: **UQ-PINN**. An overview of the PINN architecture applied to the Poisson equation. The inputs are $x$ for location and $t$ for time. The feedforward neural network produces an estimate $\hat{u}(x, t)$ of the PDE solution $u(x, t)$, as well as the estimated standard deviations $\hat{\sigma}_f(x, t)$ and $\hat{\sigma}_b(x, t)$. In the PDE block, the left-hand side of (1) is obtained by automatic differentiation, which is compared with the residual of the PDE and combined with the standard deviation $\hat{\sigma}_f(x, t)$ to contribute to the loss $L_f$. In the boundary condition block, the mean estimate is compared with measurements at the boundary conditions and combined with standard deviation $\hat{\sigma}_b(x, t)$ to contribute to the loss $L_b$.

PINNs typically use the mean squared error loss function to estimate the solution $u$ of the PDE and are therefore not able to estimate the standard deviations of the noise, $\sigma_f$ and $\sigma_b$. Instead, we utilized the mean negative Gaussian log-likelihood (NLL) as a loss function to enable estimating the standard deviation. Combining the NLL loss with additional output in the architecture enables UQ-PINN to estimate the mean and standard deviation of the data.

The loss function $L = L_f + L_b$ combines individual losses $L_f$ for noisy observations of the source term and $L_b$ for the boundary conditions. In effect, $L_f$ acts as a regularizer that penalizes solutions that do not adhere to the PDE dynamics, and $L_b$ enforces the consistency with the boundary conditions. These two individual losses are defined as

$$L_f = \frac{1}{N_f S_f} \sum_{i=1}^{N_f S_f} \text{NLL}(\tilde{f}_i; \mathcal{F}(\hat{u}(x_i^{(f)}), \hat{\sigma}_{(f)}^2(x_i^{(f)})) \,, \quad L_b = \frac{1}{N_b S_b} \sum_{i=1}^{N_b S_b} \text{NLL}(\tilde{b}_i; \hat{u}(x_i^{(b)}), \hat{\sigma}_{(b)}^2(x_i^{(b)})),$$

with $\hat{u}$ the mean prediction and $\hat{\sigma}_{(\cdot)}^2$ the estimated standard deviation.

## 3 EXPERIMENTS

**Data**. To validate our UQ-PINN method, we consider the Poisson equation $-\frac{\partial^2 u}{\partial x^2} = \pi^2 \sin(\pi x)$, with boundary conditions $u(x) = 0$, $x \in \{-1, 1\}$, and analytical solution $u(x) = \sin(\pi x)$ in the domain $\Omega = [-1, 1]$. $D_f$ has 16 uniformly sampled sensor locations with $S_f = 5$ samples per domain point. $D_b$ contains of two boundary points, $x \in \{-1, 1\}$, which are sampled $S_b = 5$ times. For performance evaluation, a test data set with $N_t = 1000$ uniformly sampled points was generated in the Poisson equation domain and the exact solution was calculated. Table 1 summarizes the variances of the noise added to the data sets.

| Noise type | $\sigma_f(x)$ | $\sigma_b(x = -1)$ | $\sigma_b(x = 1)$ |
|---|---|---|---|
| Homoscedastic | $5.0 \times 10^{-2}$ | $5.0 \times 10^{-2}$ | $5.0 \times 10^{-2}$ |
| Linear | $5.0 \times 10^{-2} \times (x + 1) \times N_f$ | $5.0 \times 10^{-2}$ | $5.0 \times 10^{-1}$ |
| Periodic | $\frac{1}{8}\left|\sin(2\pi x / \max(x))\right|$ | $5.0 \times 10^{-2}$ | $5.0 \times 10^{-1}$ |

Table 1: **Different types of generated noise added in the data sets.**

**Experimental setup and training details.** The PINN architectures consist of 3 hidden layers with 50 neurons per layer initialized using the uniform Glorot distribution. The network was trained for 100,000 epochs using the ADAM optimizer with a learning rate of $1 \times 10^{-3}$. The $\hat{\sigma}$ networks of the UQ-PINN consist of 3 layers with 10 neurons per layer. The weights of the subnetworks are frozen for half of the number of epochs. After half the number of epochs, the weights are unfrozen and the $\hat{\sigma}$ networks are updated together with the mean estimation network.

**Evaluation metrics**. We use various metrics to compare the performance of the different approaches. All considered architectures, except vanilla PINN, also output an estimate of the standard deviations $\hat{\sigma}_{(b)}$ and $\hat{\sigma}_{(f)}$, which correspond to the standard deviations of noise at the boundary conditions and of the noise for the source term. We measure the quality of our estimates using the Jensen-Shannon divergence, which is a symmetric distance measure between two distributions $P$ and $Q$ Fuglede & Topsoe (2004):

$$JS(P||Q) = \frac{1}{2}KL\left(P \left\| \frac{P+Q}{2}\right.\right) + \frac{1}{2}KL\left(Q \left\| \frac{P+Q}{2}\right.\right), \tag{3}$$

To evaluate the ability of the different methods to estimate the PDE solution, we use the relative error $L2$ calculated at all test points:

$$L2 = \frac{1}{N_{test}} \frac{\|u - \hat{u}\|_2^2}{\|u\|_2^2} \,,$$

where $u$ is the solution the PDE and $\hat{u}$ is the predicted solution with PINNs.

## 4 RESULTS

Figure 2 shows examples of the estimated solution for the Poisson equation and standard deviations of the noise using the split-hidden-unit network. We observe that our approach is capable of estimating accurate solutions while also estimating the correct pattern of the standard deviation of the noise.

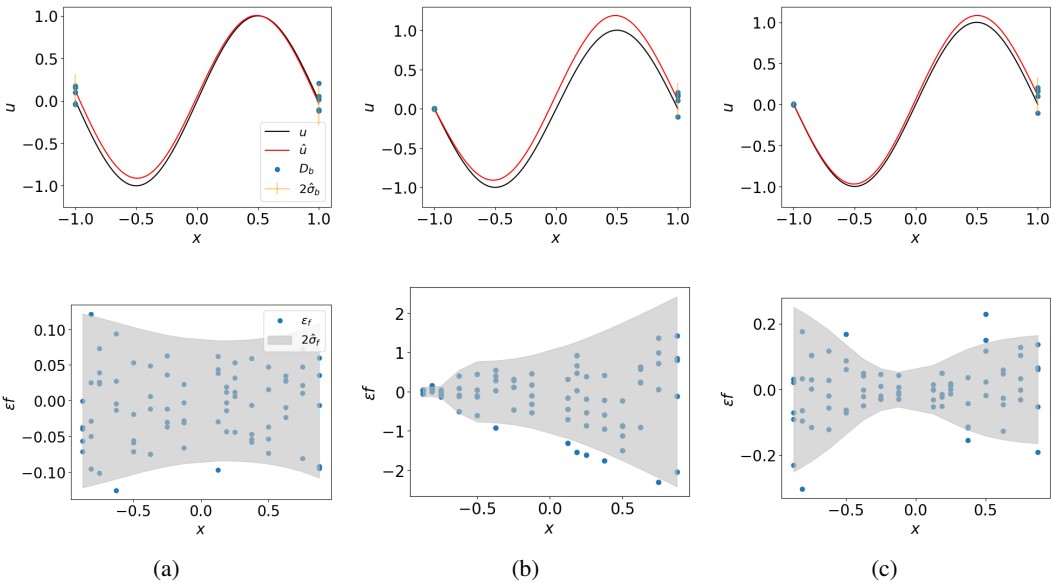

Figure 2: **Estimated UQ-PINN solution.** The first row gives an example of the sampled data points used to train the PINNs and the predictions, $\hat{u}$, of the UQ-PINN. The second row illustrates the noise in the source term $\epsilon_f$ and the two estimated standard deviations $2\hat{\sigma}_f$. Different columns correspond to different types of noise: (a) homoscedastic, (b) linearly increasing, and (c) periodic.

Table 2 presents the Jensen-Shannon divergence (JS PDE for the source term and JS BC for the boundary conditions), which measures the difference between the original and the estimated distribution. We observe that the UQ-PINN methods outperform the constant method in terms of JS-divergence for both estimation of the noise added to the boundary conditions and the source term. The only exception is homoscedastic noise, where the PINN with constant variable architecture slightly outperforms the UQ-PINN in terms of JS-divergence for the source term. This is not surprising, since the constant-variable architecture has a homoscedastic noise assumption due to having a single constant to estimate noise.

| Method | Noise | Mean residual | Mean relative L2 | JS PDE | JS BC |
|---|---|---:|---:|---:|---:|
| Split | Homoscedastic | $\mathbf{6.44 \times 10^{-3}}$ | $2.07 \times 10^{-2}$ | $5.50 \times 10^{-2}$ | $\mathbf{1.50 \times 10^{-1}}$ |
| Constant | Homoscedastic | $7.62 \times 10^{-3}$ | $2.10 \times 10^{-2}$ | $\mathbf{3.47 \times 10^{-2}}$ | $1.70 \times 10^{-1}$ |
| Vanilla | Homoscedastic | $7.93 \times 10^{-3}$ | $\mathbf{2.01 \times 10^{-2}}$ | - | - |
| Split | Linear | $4.85 \times 10^{-2}$ | $2.37 \times 10^{-1}$ | $\mathbf{1.21 \times 10^{-1}}$ | $\mathbf{1.49 \times 10^{-1}}$ |
| Constant | Linear | $3.70 \times 10^{-2}$ | $2.22 \times 10^{-1}$ | $1.82 \times 10^{-1}$ | $5.46 \times 10^{-1}$ |
| Vanilla | Linear | $\mathbf{1.89 \times 10^{-2}}$ | $\mathbf{2.20 \times 10^{-1}}$ | - | - |
| Split | Periodic | $1.03 \times 10^{-2}$ | $\mathbf{2.21 \times 10^{-1}}$ | $\mathbf{1.49 \times 10^{-1}}$ | $\mathbf{6.64 \times 10^{-2}}$ |
| Constant | Periodic | $1.54 \times 10^{-2}$ | $2.23 \times 10^{-1}$ | $1.82 \times 10^{-1}$ | $1.16 \times 10^{-1}$ |
| Vanilla | Periodic | $1.68 \times 10^{-2}$ | $2.22 \times 10^{-1}$ | - | - |

Table 2: **Performance of PINN methods estimating the Poisson equation with noise.** The best results are highlighted in bold. Results are averaged over 25 runs with different seeds. Two baselines are considered: PINN with a constant parameter for the standard deviation of the noise (Xiang et al., 2021) and vanilla PINN (Raissi et al., 2017).

## 5 CONCLUSION

In this paper, we provide a computationally inexpensive approach for quantification of aleatoric uncertainty in PINNs called UQ-PINN. In the UQ-PINN architectures, we extend the PINN by incorporating additional output. These outputs enabled the network to learn a function that approximates the standard deviation of noise in the datasets with respect to the input. We used the negative Gaussian log-likelihood loss function for the proposed UQ-PINN. We compared the UQ-PINN architecture with vanilla PINN and PINN with an external constant to estimate the standard deviation of the noise in data. The comparison was made in terms of the quality of the approximation of the PDE solution using $L2$ score and JS-divergence in terms of the noise variance estimation accuracy.

The UQ-PINN performed similarly to other PINN architectures in terms of mean error $L2$. However, the vanilla method is unable to approximate noise, and the constant architecture can only estimate homoscedastic noise. We found that the constant architecture demonstrates performance comparable to that of UQ-PINN in scenarios involving homoscedastic noise or a limited number of data samples. In the case of heteroscedastic noise, the UQ-PINN showed a performance superior to that of constant-variable PINN. Both in terms of the estimation of the mean solution and standard deviation of the noise.

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
