# OpenReview forum: "Heteroscedastic uncertainty quantification in Physics-Informed Neural Networks"
_ICLR.cc/2024/Workshop/AI4DiffEqtnsInSci — AI4DiffEqtnsInSci @ ICLR 2024 Poster_

### Official Review · Reviewer_ZQJy · 2024-02-23
**Review for UQ-PINN's**

**Rating:** 5
**Confidence:** 5

**Review:**

The authors have introduced UQ-PINN's for uncertainty quantification of the inherent noise present in the system. They have introduced a simple framework by adding multiple outputs to the PINN and modifying the regular MSE loss function with negative loss likelihood.

The method is not exactly novel and is widely used in traditional probabilistic deep learning community.
The example shown in the paper is quite simple and straightforward. It might be better if more complicated examples with PDE's are shown. It is natural since PINN's were first introduced to solve PDE's.

---

### Official Review · Reviewer_RsV1 · 2024-02-26
**This work presents a noise guided analysis for Physics-informed neural networks (PINNs) to solve differential equation problems.**

**Rating:** 7
**Confidence:** 3

**Review:**

This work presents a noise guided analysis for Physics-informed neural networks (PINNs) to solve differential equation problems.
The paper is well written and some conclusions on the performance of the UQ-PINN are demonstrated on the Poisson equation with additive noise.

---

### Meta-Review · Area_Chair_k1qZ · 2024-02-26

**Recommendation:** Accept (Poster)

**Metareview:**

Both of the reviewers acknowledge the contribution of adding UQ to Physics-Informed Neural Networks. UQ is very relevant to the workshop and needed in SciML. I recommend the authors expand the literature review and references, e.g., Zou et al. "NeuralUQ: A comprehensive library for uncertainty quantification in neural differential equations and operators" (https://arxiv.org/abs/2208.11866) and benchmarking against other UQ approaches, such as conformal prediction. I vote for acceptance for the workshop.

---

### Decision · Program_Chairs · 2024-02-28

Accept (Poster)